# Themes, communities and influencers of online probiotics chatter: A retrospective analysis from 2009-2017

Santosh Vijaykumar[1]*, Aravind Sesagiri Raamkumar[2], Kristofor McCarty[1], Cuthbert Mutumbwa[3‡], Jawwad Mustafa[1‡], Cyndy Au[4]

1 Department of Psychology, Faculty of Health & Life Sciences, Northumbria University, Newcastle upon Tyne, United Kingdom, 2 Saw Swee Hock School of Public Health, National University of Singapore, Singapore, Singapore, 3 Department of Computer and Information Sciences, Northumbria University, Newcastle upon Tyne, United Kingdom, 4 Kong Chian School of Business, Singapore Management University, Singapore, Singapore

☯ These authors contributed equally to this work.
‡ These authors also contributed equally to this work.
* santosh.vijaykumar@northumbria.ac.uk

**Data Availability Statement:** Our data have now been made publicly available and can be access via this link: https://osf.io/756jw/.

## Abstract

We build on recent examinations questioning the quality of online information about probiotic products by studying the themes of content, detecting virtual communities and identifying key influencers in social media using data science techniques. We conducted topic modelling (n = 36,715 tweets) and longitudinal social network analysis (n = 17,834 tweets) of probiotic chatter on Twitter from 2009–17. We used Latent Dirichlet Allocation (LDA) to build the topic models and network analysis tool Gephi for building yearly graphs. We identified the top 10 topics of probiotics-related communication on Twitter and a constant rise in communication activity. However the number of communities grew consistently to peak in 2014 before dipping and levelling off by 2017. While several probiotics industry actors appeared and disappeared during this period, the influence of one specific actor rose from a hub initially to an authority in the latter years. With multi-brand advertising and probiotics promotions mostly occupying the Twitter chatter, scientists, journalists, or policymakers exerted minimal influence in these communities. Consistent with previous research, we find that probiotics-related content on social media veers towards promotions and benefits. Probiotic industry actors maintain consistent presence on Twitter while transitioning from hubs to authorities over time; scientific entities assume an authoritative role without much engagement. The involvement of scientific, journalistic or regulatory stakeholders will help create a balanced informational environment surrounding probiotic products.

## Introduction

Probiotics are defined as live microorganisms that confer a health benefit upon the host when administered in adequate amounts [1]. Scientific evidence demonstrating its positive health

**Funding:** This project was funded by the Consumer Data Research Centre, an ESRC Data Investment, under project ID CDRC 085, ES/L011840/1; ES/L011891/1. The grant was awarded to SV (PI) and KM (Co-I). The funders had no role in study design, data collection and analysis, decision to publish, or preparation of the manuscript. CA is employed with International Foods and Flavours (IFF) but participated in this study in the capacity of their academic affiliation with Singapore Management University. The funder (IFF) provided support in the form of salaries for CA, but did not have any additional role in the study design, data collection and analysis, decision to publish, or preparation of the manuscript. The specific roles of these authors are articulated in the 'author contributions' section.

**Competing interests:** CA is employed with IFF. This does not alter our adherence to PLOS ONE policies on sharing data and materials.

effects has however been inconclusive [2]. Because probiotics might be beneficial to individuals with specific health conditions as opposed to the general population, some regulatory agencies like the European Food Safety Authority (EFSA) have ruled against manufacturers displaying claims about the health benefits on probiotic product labels [3, 4].

Curiously, neither the equivocal nature of scientific evidence casting a shadow on the health benefits of probiotic products nor the accompanying labelling controversies, have stemmed the growing popularity of probiotic products among consumers. Instead, the probiotics industry is predicted to grow from \$35.6bn in 2015 to \$64.6bn by 2023 [5–8]. In the UK, Google searchers for the term "probiotic" have doubled over the past five years [3].

One of the explanations for this paradox lies in marketing strategies employed by the probiotic industry. Historically, the probiotics industry has gained growth through traditional advertising [9], but in the last decade the conversation has expectedly shifted to online channels. This development has however created problems. While research on digital probiotics content is relatively scant, an examination of online probiotics messages found an overwhelming promotion of the benefits of probiotics [10]. A recent study of the top 150 probiotics web pages listed by Google revealed a vast majority hosted by commercial enterprises providing the least reliable information containing claims mostly unsupported by scientific evidence [11]. Probiotics claims are also appearing on social media [12], a virtual, networked crucible of multiple individuals and communities that can communicate, produce and share content. There are thus two interlinked aspects of virtual probiotics communities–content dynamics and community dynamics–that command our attention.

One of the ways to analyse the dynamics of online probiotic content is by examining the latent structures of conversational themes or topics that underlie social media chatter on platforms like Twitter. Such analysis is facilitated by topic modelling, a data-intensive automated approach to content analysis that is being increasingly used to examine social media chatter related to a range of health-related issues. For instance, Franz et al. (2019) analysed textual corpuses from online forums related to self-injurious thoughts and behaviours from online blogs and identified specific themes including suicide ideation, depression and abuse which characterized these discussions [13]. Other researchers suggest that topic modelling could be used to detect vaccine safety signals from social media data as an alternative, proactive strategy to measure vaccine-related sentiments [14]. These studies highlighted how insights gained through topic modelling could contribute to the design and conceptualization of public health interventions and inform the methodological rationale for our work. However, our study builds on work related to analysis of social media conversations related to HPV vaccines by Surian and colleagues (2016) that combine an examination of underlying conversational themes with detection of online communities premised on the rationale that the former shape the latter [15]. Applied to the context of this study, this approach will allow us to understand how the specific topics underlying online probiotics conversations might be situated in the larger context of online probiotics communities.

The evolution of probiotics chatter on Twitter can be understood through the lens of viral marketing–a type of marketing strategy where information about a product or service spreads through word-of-mouth on online social media networks [16]. This process of information diffusion triggers communication between individual or groups of consumers, usually between organisations and consumers [17]. Given that social media platforms such as Twitter enables the co-existence of a range of actors in the nutritional ecosystem [18], interactions about probiotics could also occur between either of these two actors and other ancilliary stakeholders that are related to the product in question. In the probiotics context, these stakeholders could include academics who study probiotics [19], policymakers involved in its regulation [20], retailers involved in selling probiotic products [21], fitness or sports-related individuals or

professionals [22], and dietetics professionals who could use social media as a professional information resource [23].

The series of interactions leads to the gradual evolution of online communities comprising social media users with similar characteristics who engage in information seeking and sharing, with more engagement leading to greater social capital [24]. Social network theory (SNT) allows us to examine the structure of these communities and identify which actors or entities might be central (hubs) or peripheral to the network, along with understanding their evolution, growth and decline over time [25]. Further, SNT allows us to understand the communicative behaviours of these entities (in-degree vs. outdegree). In the context of the probiotics chatter on Twitter, for instance, we can identify if specific types of actors (e.g. retailers or policymakers) in the probiotics industry were actively communicating to other members in the network (measured with out-degree) or whether they were being communicated to (measured with in-degree). These constructs could be used to measure the actor's influence in the network.

The concept of influence is especially relevant to online communities as conversations on social media are increasingly being driven by a host of social media "influencers". Young (2019) found that various consumer probiotics drinks companies used influencers for their marketing campaigns, including dieticians, nutritionists and bloggers [12]. The dialogue around probiotics has been pushed by other social media figures, such as health practitioners, sport personalities, athletes and marketers [26]. Influencer marketing has been growing on social media since the early days of Web 2.0, however, has been widely adopted in more recent years [27]. The key ingredients of successful influencer marketing figures–authenticity, credibility and perceived closeness–are present in current social media health marketing.

While authenticity and perceived closeness are important, expertise is a big part of building credibility in health communication. Gillin (2009) explains that social media influencers are those that are experts in their fields, stating that they can be researchers and practitioners just as much as it could be people with lived experience of the issue and product [28]. Raafat (2018) analysed social media content of health experts and those with lived experience and found that consumers trusted both [26]. Authenticity in the experience of the health issue or product supplanted established and officially recognised expertise. The non-expert health influencers used their lived experience to embody the idea of authenticity to what they were saying about health-related issues. The personalisation of their experiences to form a bond with their audiences was key to maintain their influence. Nichols (2017) notes that deciding expertise on a platform where anyone can claim it makes it a challenge to sort the information based in science in comparison to faux claims [29].

Influencers rely on wellbeing and health to present themselves as aspirational. Their platforms stray into the health domain [30]. This can often lead to misinformation. Social media influencers tended to recommend and portray types of diets that were not necessary to their followers, having no expertise in the area [31].

In summary, analysing the conversation around probiotics on social media will enable public health professionals including nutritionists, dieticians, food safety agencies and scientists to understand the dynamics of online probiotics information environment. Twitter, a popular microblogging platform populated by several of these stakeholders has been previously studied to understand people's food consumption behaviours [32], consumers' depiction of health maintenance behaviours [33], and interactions between food agencies and communities [20]. The aim of this study is to extend this line of research to understand the latent nature of conversation themes around online probiotics chatter and examine the nature and lifespan of probiotic communities on Twitter by analyzing a longitudinal dataset of probiotic-related tweets in the United Kingdom. The study makes use of topic modelling to identify the prevalent

probiotic topics while social network analysis study techniques are used to analyze data pertaining to probiotic conversations in Twitter.

## Research questions

RQ1: What are the top 10 topics that have characterised Twitter conversations around probiotic products in the United Kingdom from 2009–2017?

RQ2: What are the online communities that have engaged in Twitter discourse surrounding probiotics? How have the probiotic online communities changed over time?

RQ3: Which Twitter accounts have emerged as hubs and authorities in Twitter chatter related to probiotics? How has their role changed over time?

RQ4: Do probiotic Twitter accounts post tweets or get tagged in tweets at a predominant level? How have these social media conversation dynamcs related to probiotic products changed over time?

# Materials and methods

## Data extraction

We first identified all tweets containing the term 'probiotic' or 'probiotics' between 16[th] May 2009 and 30[th] May 2017 using the social media listening platform Crimson Hexagon (CH). CH delivered tweet metadata (e.g., location, date, tweet URL) in JSON format. We then used the BeautifulSoup (web scraping), re (regular expressions), hashlib (hashing), and requests (URLretrieval) packages in Python 3.7 to retrieve each tweet, identify and anonymise (hash) Twitter handles as per our ethical obligations, and format each tweet into a readable format (.csv). This resulted in a total of 79,694 tweets, of which there were 36,715 unique/original tweets. A common concept on Twitter are 'retweets', where users can share posts that other users have created to their followers.

## Topic modeling

**Data preparation.** Topic modelling is a probabilistic statistical text-mining technique for discovering latent 'topics' within a corpora of documents (somewhat akin to dimension reduction techniques such as Principal Component Analysis or PCA). For this study, we wished to model the semantic structures within the social media conversation on Twitter surrounding probiotics. In the next stage we employed the most common topic modelling technique Latent Dirichlet Allocation (LDA) using the implementation found in the gensim package in Python [34]. This first step in this process was to remove a) web links, and twitter handles (which were previously hashed out for anonymity purposes), b) punctuation, and c) common words (e.g., 'and', 'but', 'if'), known as 'stopwords'. We used the standard US-English stopwords provided by the Natural Language ToolKit (NLTK) package. Using gensim, a natural language processing package, we then created bigrams to ensure common word couplets were kept in the model as one entity (for example 'systematic review' would combine to systematic_review), and retained only nouns, adjectives, verbs, adverbs in each tweet. Next, we Lemmatized each token (i.e., find the root word) in each tweet so that similar tokens will be recognised as the same. An example of this would be: health, healthiness, and healthy, should all be recognised as simply 'health'. Based on the rationale that words that appear too regularly are unlikely to be meaningful in topics, and words that appear too sparsely introduce noise, we filtered the words to discard any words that appeared in >80% of tweets, as well as words that appeared in <30 tweets.

## Social netwotk analysis

**Data preparation.**　In Twitter terminology, *mention* is an instance of tagging/mentioning another Twitter user in a tweet. For example, if *user A* wants to start a discussion with *user B*, the "@" character is used to tag *user B* in the tweet. It should also be noted that when a user retweets the tweet of another user, Twitter automatically adds the characters "RT @user_account" in the tweet. Hence, mentions are naturally present in retweets. From the full extract of tweets from 1$^{st}$ June 2009 to 31$^{st}$ December 2017 (N = 70,828), only the tweets containing mentions were selected. Through this process, 17,834 tweets were identified for the study.

**Twitter mentions and conversations.**　From the filtered tweets, the Twitter account name (Twitter handle) and the mentions data were extracted. A combination of Twitter account name and mention is usually referred to as a *conversation*. It is to be noted that a single tweet could contain multiple mentions.

**Communication graphs.**　The network analysis tool Gephi was used for building the graphs for each year [35]. In these graphs, the source node is the Twitter user account while the target node is the mention. This type of graph is referred to as a *directed graph* since the direction of communication is from the source to the target. After the data was loaded in Gephi for each year separately, the giant component setting was used to remove unconnected nodes in the year graph. A giant component is a connected component of a network that contains a significant proportion of the entire nodes in the network. Typically, as the network expands the giant component will continue to have a significant fraction of the nodes [36]. The giant component graphs were considered as the final set of graphs for the data analyses. Using Gephi's modularity feature, nodes were classified into different modular classes in all the year graphs. This feature is based on the Louvain method for calculating modularity [37]. A total of 53 communities were identified in the nine year time period. The Fruchterman and Reingold algorithm was used in Gephi to set the layout of the graphs [38]. The nodes in the graphs were sized based on their degree values.

**Anonymisation.**　Identification of actors facilitates the performance of and interpretation of findings from social network analysis. However, current best practices in research involving social media data (including Twitter datasets) recommend the anonymisation of users identified in the analyses due to ethical considerations [39]. In order to reconcile this paradox, we anonymised all Twitter handles that were included in our final set of findings. We first assigned exclusive user IDs (U1, U2. . .) to the users originally identified in Table 4 and found N = 57 exlcusive Twitter IDs. We then extracted each of their bios from their Twitter pages and performed two rounds of categorisation. In Round 1, we classified them as individuals (n = 21) or organisations (n = 25), accounts that ceased to exist (n = 9) and those without a bio (n = 2). Our review of the bios revealed that they could be further classified into discreet categories. Specifically, individuals were categorised as either academic (AC = 4) or non-academic users (NAC = 14); and organisations into commercial (COM = 17), media (MDA = 6), professional associations (AN = 2) and non-profits (NPR = 1). Users who could not be assigned to any of these categories were classed as others (OT = 4). Following this categorisation, users were renamed sequentially by their assigned category (e.g. COM1, COM2, etc.). To ensure consistency of the categorisation scheme, two authors coded 10% of the sample (N = 6) and found an initial agreement of 66.67% but achieved 100% agreement on a different sample of six tweets after clarifying the category descriptions.

In Table 1, statistics related to the graphs/networks generated with the tweets are listed along with the communities count. Over time, we observe an increase in the number of nodes and edges in the network indicating consistently expanding communication activities around probiotics on Twitter. In terms of average degree of nodes per year, the graphs fall into two categories (until and after 2013). Until 2013, the average degree was below 2.5. In the latter category, the average degree seems to have increased with values getting close to 3. The average

**Table 1. Measures for base graphs depicting growth of probiotics communities on Twitter from 2009–17.**

| Year | Tweets | Base Graph | | Giant Component Graph | | | | |
|---|---|---|---|---|---|---|---|---|
| | | Nodes | Edges | Nodes | Edges | Avg Degree (SD) | Avg Closeness Centrality (SD) | Communities |
| 2009 | 82 | 112 | 72 | 13 | 16 | 2.46 (2.82) | 0.29 (0.35) | 2 |
| 2010 | 620 | 618 | 484 | 209 | 232 | 2.22 (7.32) | 0.24 (0.37) | 4 |
| 2011 | 833 | 719 | 611 | 204 | 244 | 2.39 (13.86) | 0.26 (0.32) | 4 |
| 2012 | 1726 | 1558 | 1331 | 569 | 704 | 2.47 (9.01) | 0.37 (0.43) | 6 |
| 2013 | 2921 | 2444 | 2220 | 989 | 1206 | 2.43 (7.94) | 0.41 (0.39) | 8 |
| 2014 | 3662 | 2304 | 2473 | 976 | 1476 | 3.02 (14.79) | 0.44 (0.37) | 9 |
| 2015 | 2198 | 1799 | 1707 | 586 | 778 | 2.65 (5.36) | 0.46 (0.43) | 7 |
| 2016 | 2167 | 1477 | 1489 | 585 | 789 | 2.69 (4.45) | 0.5 (0.43) | 6 |
| 2017 | 3625 | 2174 | 2554 | 1219 | 1753 | 2.87 (5.97) | 0.52 (0.43) | 7 |

closeness centrality [40] per node, has been included in Table 1. This metric defines the importance of a node in the graph, by measuring how close the node is to other nodes in the graph (sum of geodesic distance between the particular node and all other nodes in the graph). In a graph of multiple nodes, the nodes with relatively lower closeness centrality values, are considered to be closer to the other nodes in the graph. With 2009 as the exception, it is observed that this metric has consistently increased every year at an average level. Although probiotic Twitter users seem to be posting more tweets through the years, the proximity to each other has been steadily decreasing as indicated by the rise in average closeness centrality values. We also notice that the probiotics network initially starts with only two communities in 2009 but grows to nine by 2014 and settles down at six to seven by 2017.

## Data analysis

**Model tuning to identify top 10 topics (RQ1).** Coherence is a common metric when evaluating the quality of topic models, and we used this to guide our final model. We ran LDA models using all combinations of the following hyper parameters:

Alpha values: 0.01, 0.21, 0.41, 0.61, 0.81, asymmetric, symmetric

Beta values: 0.01, 0.21, 0.41, 0.61, 0.81, symmetric

Topics: betweeen 1 and 20

We initially implemented the Mallet LDA model [41] via the MalletLda procedure in gensim, but this yielded quite low coherence metrics (max ~.3). We then ran 840 iterations of the LdaMulticore procedure in gensim. Table 2 summarises the top ten models and their parameters, along with their coherence score.

Coherence metrics alone do not necessarily equate to the most meaningful models. We see in Table 2 that models with between 19 and 20 topics have the highest coherence scores. However, upon inspection of these models many of these additional topics are very closely related and 'stacked' on one another in a way that does not make a lot of semantic sense. These models are ultimately too fine grain for the problem we are studying, hence, we opted for a ten topic model that strikes the balance between meaningful clustering, and coherence ($C\_v = 0.57$).

**Identification of communities (RQ2).** As mentioned earlier, 53 communities were identified in the Twitter graphs built for the nine-year time period. The prevalent theme of community was identified using a process involving one principal coder and one reviewer (two of the authors). First, the principal coder reviewed the contents of randomly selected tweets from each community and assigned the relevant theme names [42]. Next, another author reviewed and

**Table 2. The top ten models and their hyperparameters based on coherence scores.**

| Topics | Alpha | Beta | Coherence |
|---|---|---|---|
| 20 | asymmetric | 0.61 | 0.61 |
| 19 | asymmetric | 0.61 | 0.59 |
| 20 | asymmetric | 0.81 | 0.59 |
| 14 | asymmetric | 0.81 | 0.58 |
| 18 | asymmetric | 0.61 | 0.58 |
| 14 | asymmetric | 0.61 | 0.58 |
| 17 | asymmetric | 0.61 | 0.57 |
| 20 | asymmetric | 0.41 | 0.57 |
| 17 | asymmetric | 0.81 | 0.57 |
| 10 | asymmetric | 0.81 | 0.57 |

confirmed the themes assigned by the principal coder. It is to be noted that this author did not independently assign the themes to the tweets, rather reviewed the assignments of the first coder. If a community had more than one major theme, the community's label was set to *theme1 and theme2*. For each of these themes, the number of constituent communities are reported. We did not use any codebook for this exercise.

**Authorities and hubs (RQ3).** In graph/network theory, the in-degree of a node (Twitter account in the context of this study) is the number of incoming edges to that particular node. Similarly, the out-degree of a node is the number of outgoing edges from the particular node. In this study's context, edge refers to a tweet where the tweeting user is the source node and the tagged user in the tweet is the target node. Twitter accounts with high in-degree values are considered as authorities since these accounts get tagged more in tweets. On the other hand, Twitter accounts with high out-degree values are considered as hubs since these Twitter accounts tweet more about probiotics. We rank the Twitter accounts based on the out-degree and in-degree values for identifying authorities and hubs respectively. We considered the top 5 ranks for our analysis in this study.

**Posting and tagging behavior correlation (RQ4).** For this analysis, we plotted the in-degree values against the out-degree values of accounts for each year using scatter plots. We interpreted the findings by observing the grouping of the accounts towards a particular axis. For instance, if the accounts were more closer to the y-axis in the plot, it can be inferred that such accounts post more tweets in contrast to tweets where they are tagged. In addition, we calculated the percentage of accounts which had a higher out-degree value than in-degree value for each year. This percentage helps in identifying whether posting or being tagged was the predominant activity for a particular year. For instance, if the percentage was above 50%, it means there are more accounts with a higher out-degree value than in-degree value. Hence, posting behavior can be considered to the dominant activity in the network for that year. This analysis pertains to RQ2.

# Results

## Top 10 probiotic topics (RQ1)

Fig 1 graphically represents the top 10 words in each of the top ten topics. The size of each word increases as a function of it's relative frequency in the model. Topic modelling is similar to PCA in terms of its output, whereby the model shows us the most prevalent words (and their weights) it has grouped together, but is ultimately up to the author to interpret what these latent structures pertain to. For example, in Topic 0 we see that "food", "good"

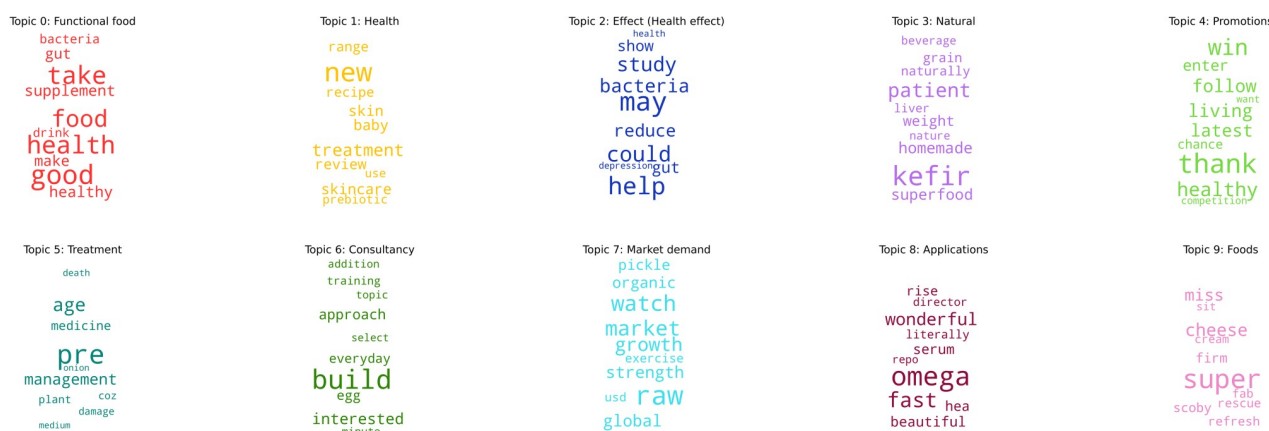

**Fig 1. A series of word clouds depicting the top 10 words in each topic.** The size of each word increases as a function of it's relative frequency in the model. The model shows us the most prevalent words (and their weights) it has grouped together.

and "health" are very prominent in this topic and we named this Functional Food. A further example would be Topic 7 that features prominent words such as "market" and "growth". We termed this topic "Market Demand". We also explored the weightings of the top 10 words in each topic and found that the word clouds in Fig 1 mostly mirror the top weighted words (a histogram of the top ten word weights relative to their frequencies can be found in S1 Fig).

In order to visualise the grouping of each document by topic, and the relative distances between topics (i.e., topic distinctiveness), we plotted a t-distributed stochastic neighbour embedding (t-SNE) plot using the Sci-Kit Learn and Bokeh packages in Python (see Fig 2). The plot was generated using a learning rate (epsilon) of 250, with a perplexity value of 30 and a step value of 5000 (iterations). The plot denotes each topic using a colour, with each point being a document (tweet). The distances between each topic indicate the inter-topic distance. We can see here that Topic 0 (Functional Food) is by far the largest and most dominant topic with the most tweets. It is also quite distinct from other topics in the array as most of the tweets are clustered together without other colours (topics) mixed in. Similary, Topic 2 (Health Effect), is the second most populous topic, and again is quite distinct from other topics in the array. Conversely, Topics 3 through 9 are smaller, more fine grained topics that are very intertwined with one another. With topics that are semantically related (e.g., Topic 4: promotions and Topic 7: market demand), we see in Fig 2 that clusters of these topics emerge together. Ultimately the model has to assign each tweet with one topic based on its weight, even if the words contained within it span multiple topics. This likely explains the clustering and overlaps in the centre of the figure. Furthermore, the figure shows a lot of the tweets form filiform structures across the 2D plane. Our interpretation of these structures is that they may form runaway conversations (replies) and occasionally switch topic part way through. Caution must be taken here as the dimension reduction used in t-SNE plots can lead to patterns that are exaggerated or misleading. However, we tried a range of perplexity values (5, 20, 30, 40, 50), step counts (1000–5000 in steps of 500) and learning rates (100, 150, 200, 250) and on each occasion, these kinds of structures and clustering emerged.

## Identification of probiotic communities on Twitter (RQ2)

Five unique community themes emerged from the list of communities that were detected in the probiotic Twitter graphs. In Table 3, the community themes are listed along with the

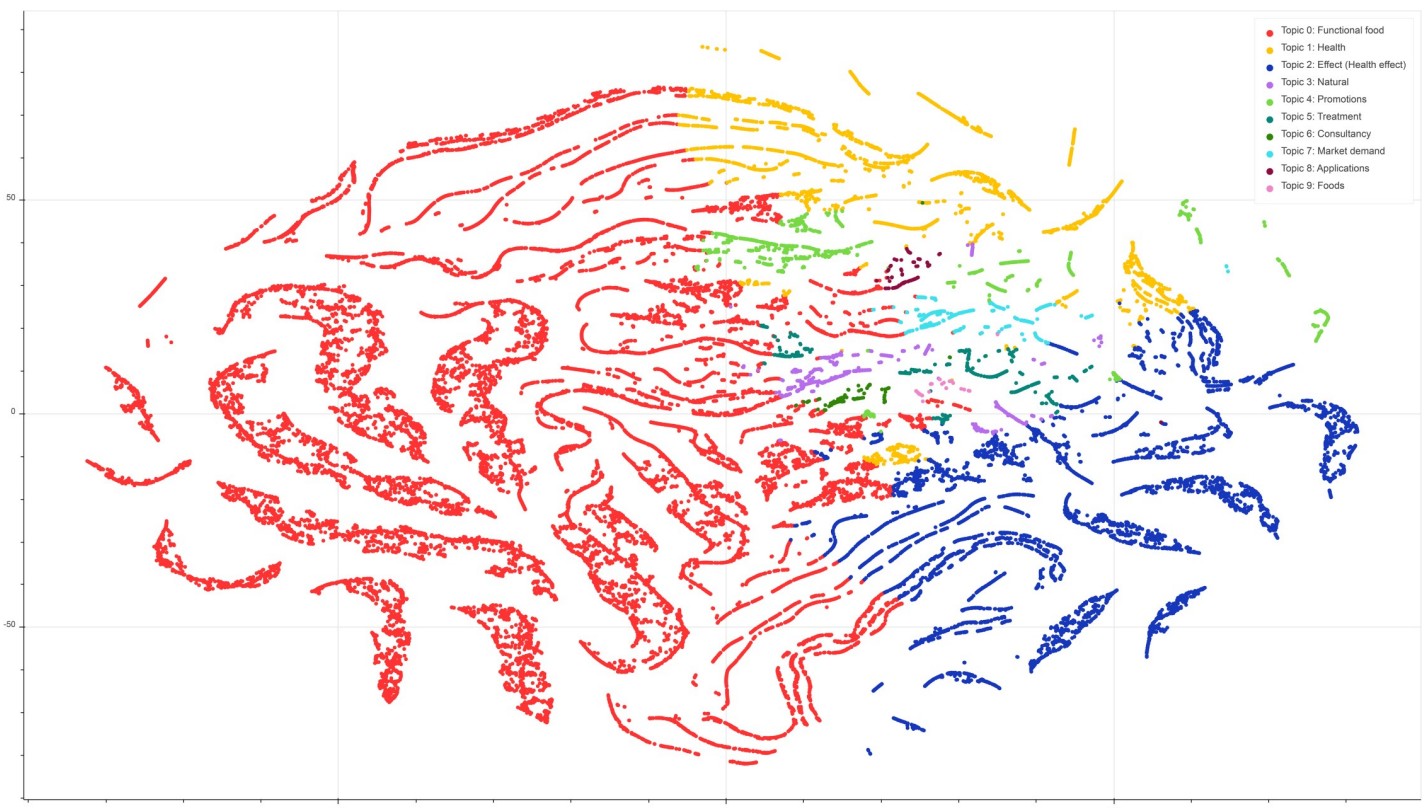

**Fig 2. t-SNE plot illustrating the distribution of each tweet, and it's dominant topic (colour).** The plot denotes each topic using a colour, with each point being a document (tweet). The distances between each topic indicate the inter-topic distance. For instance, Topic 0 (Functional Food) is by far the largest and most dominant topic with the most tweets. It is also quite distinct from other topics in the array as most of the tweets are clustered together without other colours (topics) mixed in.

descriptions and verbatim tweets. The prevalence count of the communities are also included with the community names. Health benefits of probiotics was the major theme represented in tweets with atleast 18 communities. The second most popular theme was multibrand advertising with different probiotic brands being advertised. This theme was represented by 13 communities in the nine year time period. COM1 adversiting was the third most popular theme (n = 8) in the tweets. In the communities representing this theme, the focus was specifically found to be in advertising COM1 products. There were seven communities in which the health effects of probiotics were discussed. In these communities, the frame of reference was scientific literature and grey literature on probiotics. We also found three communities in which the tweets were posted to publicize probiotics product promotions. These promotions were mostly competitions where the winners get vouchers for free probiotic products.

In Fig 3, the change in the community theme trends across the nine years are visualized in the form of an alluvial diagram. Three trends can be observed in the figure. The first trend is the consistent presence of COM1 adversiting community in all the nine years. Until 2014, this community had more Twitter accounts tweeting for it. The second trend is the increase in the prevalence of multibrand advertising community. Although, this community was first observed in 2012, the community did not have a big presence between 2012 and 2014. Since 2015, the number of accounts representing this community has consistently increased. The third trend is the presence of health benefits and health effects communities so that the discussion on probiotic effectiveness and benefits remained consistent all through the years.

Table 3. Community themes with descriptions and exemplars.

| Community Theme (n) | Description with Verbatim Tweets |
|---|---|
| Health Benefits (18) | In this community, Twitter users often mention the health benefits of using probiotics. However, the focus is not on specific products. |
| | *Tweets*: |
| | Probiotics and Fermented Foods for a Healthy Immune System http://t.co/Aacgwp7k via @NAC11 |
| | Forget Prozac—Try probiotics to ease anxiety |
| Multibrand Advertising (13) | In this community, Twitter users often mention the benefits of using probiotics products. The aim is to promote the usage of products manufactured by companies. |
| | *Tweets*: |
| | . . . yummy! do you fancy trying some delicious Belgian #probiotic #chocolate next Monday? you'll love it—DM me details 2 get samples |
| | Aurelia Probiotic Skincare introduces the Cell Revitalise Rose Mask & Eye Revitalising Duo http://t.co/Mlg0aioCOP (@NA2) |
| COM1 Advertising (8) | In this community, Twitter users specifically focus on COM1 products. They either promote the products, talk about the health benefits, or retweet competitions conducted to get free COM1 products. |
| | *Tweets*: |
| | @COM1 ok ok! #FF @COM1 for their fantastic range of probiotics to keep your gut nice and healthy this Christmas and New Year! WIN! The FIRST EVER pack of our Brand New Premium Probiotic http://bit.ly/biL0X5 - RT & Follow @COM1 to enter #competition |
| Health Effects (7) | In this community, Twitter users share scientific articles and grey literature (webpages, blog articles) on probiotics. However, the focus is not on promotion on probiotics, rather personal opinions and even criticisms on probiotics. |
| | *Tweets*: |
| | Reduce your belly fat with probiotics? http://t.co/DoZcLPB via @. . . |
| | Probiotics—What Scientific Basis Do They Have? http://t.co/pFc0HVr via @. . . |
| Promotions (3) | In this community, Twitter users share and retweet tweets in which public can win probiotics products by taking part in competitions. |
| | Tweets: |
| | RT @COM18 AND NOW FOR GOLD in our #Olympic #Skincare #Giveaway! RT&FLW to win EXCLUSIVE probiotic skincare set. http://t.co/UVSXN3ER Â. . . |
| | Enter our competition to win a bottle of @. . . .. Blend Probiotics. Simply RT b4 midnight to enter! rrp Â£20.99 #fridayfever |

## Identification of top hubs and authorities (RQ3)

In Table 4, the top five ranked probiotic Twitter hubs and authorities are listed based on their out-degree values and in-degree values respectively. Except for 2017, COM1 emerges as the top hub with the highest out-degree values consistently from 2009 to 2016. NAC2 was one of the top hubs in 2010 (out = 54), 2011 (out = 7) and 2014 (out = 13). NA2 and NAC8 briefly appear in the top 5 hub ranks between 2013 and 2015. In 2017, NAC1 (out = 34) and NAC3 (out = 30) have tweeted more about probiotics than COM1 (out = 22).

Examining in-degree values, we see that COM1 emerges as the top ranked authority in all years except 2013 and 2016. In the year 2013, COM3 (in = 172) was tagged in more tweets while in 2016, AN1 (in = 34) and NA7 (in = 31) were tagged in more tweets than COM1 (in = 30). Apart from COM1, none of the other Twitter accounts appear in the top 5 authority ranks for all the nine years. A graphical visualisation of the top ranked hubs and authorities in the form of bump charts is available in Fig 4.

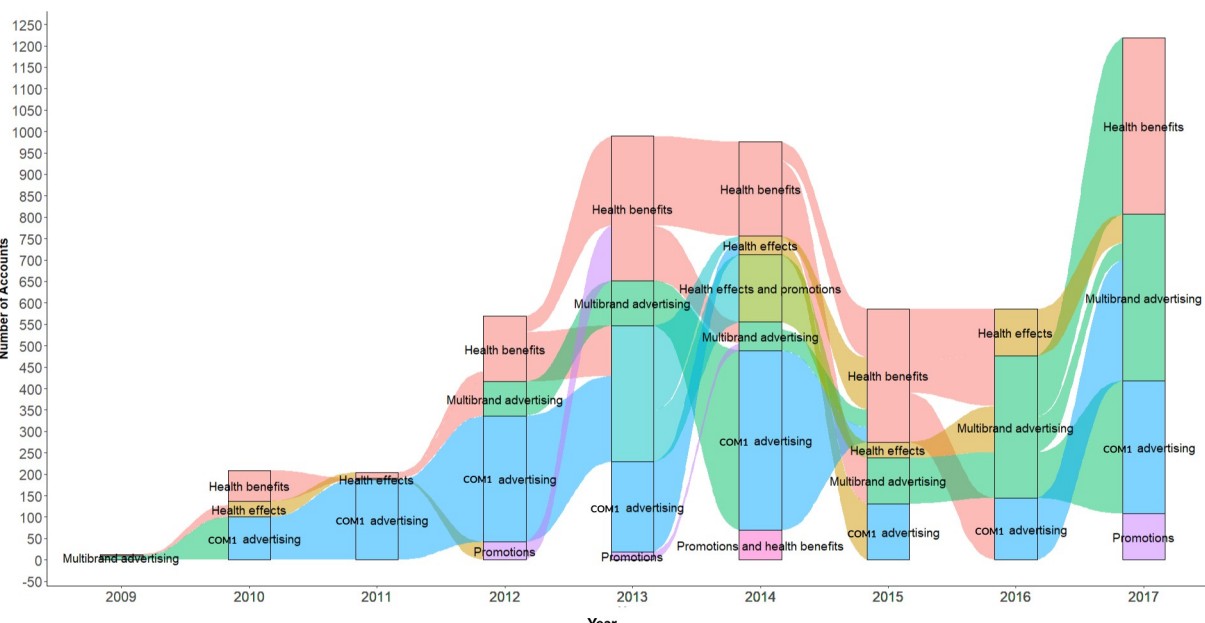

**Fig 3. Alluvial diagram showing the variations in online probiotic community themes on Twitter from 2009–2017.** Each coloured block represents a theme with the stream fields showing how the respective themes varied from one year to the next. There are total of five themes across the nine year period. All the five themes were noticed for the years 2013 an 2014. Health Benefits, COM1 Advertising and Multibrand Advertising are the most popular themes.

**Table 4. Ranking of top Twitter accounts by out-degree and in-degree frequencies used as surrogates to identify hubs and authorities respectively within the UK probiotcs network.**

| Year | Top Hubs: Twitter Accounts by Rank (out-degree) | | | | |
|---|---|---|---|---|---|
| | 1 | 2 | 3 | 4 | 5 |
| 2009 | COM1 (6) | NPR1 (4) | COM4 (2) | NAC6 (2) | COM10 (1) |
| 2010 | COM1 (54) | NAC2 (54) | NAC4 (13) | COM6 (8) | MDA4 (5) |
| 2011 | COM1 (128) | NAC2 (7) | NA1 (3) | NAC7 (2) | NAC9 (2) |
| 2012 | COM1 (126) | COM2 (54) | NAC5 (31) | NAC14 (11) | NA3 (11) |
| 2013 | COM1 (69) | MDA1 (30) | COM5 (26) | NA2 (16) | NAC10 (15) |
| 2014 | COM1 (50) | COM3 (15) | NAC2 (13) | COM8 (12) | NAC8 (10) |
| 2015 | COM1 (48) | MDA2 (23) | MDA3 (23) | NAC8 (21) | NA2 (18) |
| 2016 | COM1 (25) | AC1 (17) | MDA2 (15) | COM9 (12) | COM11 (10) |
| 2017 | NAC1 (34) | NAC3 (30) | COM1 (22) | OT2 (20) | COM12 (19) |

| Year | Top Authorities: Twitter Accounts by Rank (in-degree) | | | | |
|---|---|---|---|---|---|
| | 1 | 2 | 3 | 4 | 5 |
| 2009 | COM1 (5) | NA4 (2) | NPR1 (1) | OT4 (1) | AC4 (1) |
| 2010 | COM1 (36) | COM13 (5) | NAC11 (5) | AC2 (4) | MDA5 (4) |
| 2011 | COM1 (71) | OT1 (4) | OT3 (4) | NA8 (4) | COM17 (3) |
| 2012 | COM1 (70) | NA5 (22) | COM14 (19) | MDA5 (16) | COM18 (14) |
| 2013 | COM3 (172) | COM1 (65) | COM15 (47) | MDA5 (26) | NAC11 (24) |
| 2014 | COM1 (320) | NA6 (268) | MDA5 (25) | COM11 (20) | MDA6 (18) |
| 2015 | COM1 (42) | NA2 (34) | NAC12 (24) | COM3 (19) | COM11 (18) |
| 2016 | AN1 (34) | NA7 (31) | COM1 (30) | AC3 (27) | NA9 (25) |
| 2017 | COM1 (105) | AN1 (61) | AN2 (55) | COM16 (54) | NAC13 (50) |

AN: *Professional associations* | COM: *Commercial organisations* | NPR: *Non-profit organisations* | MDA: *News and Media organisations* | AC: *Academic individuals* | NAC: *Non-academic individuals* | NA: *Account does not exist*

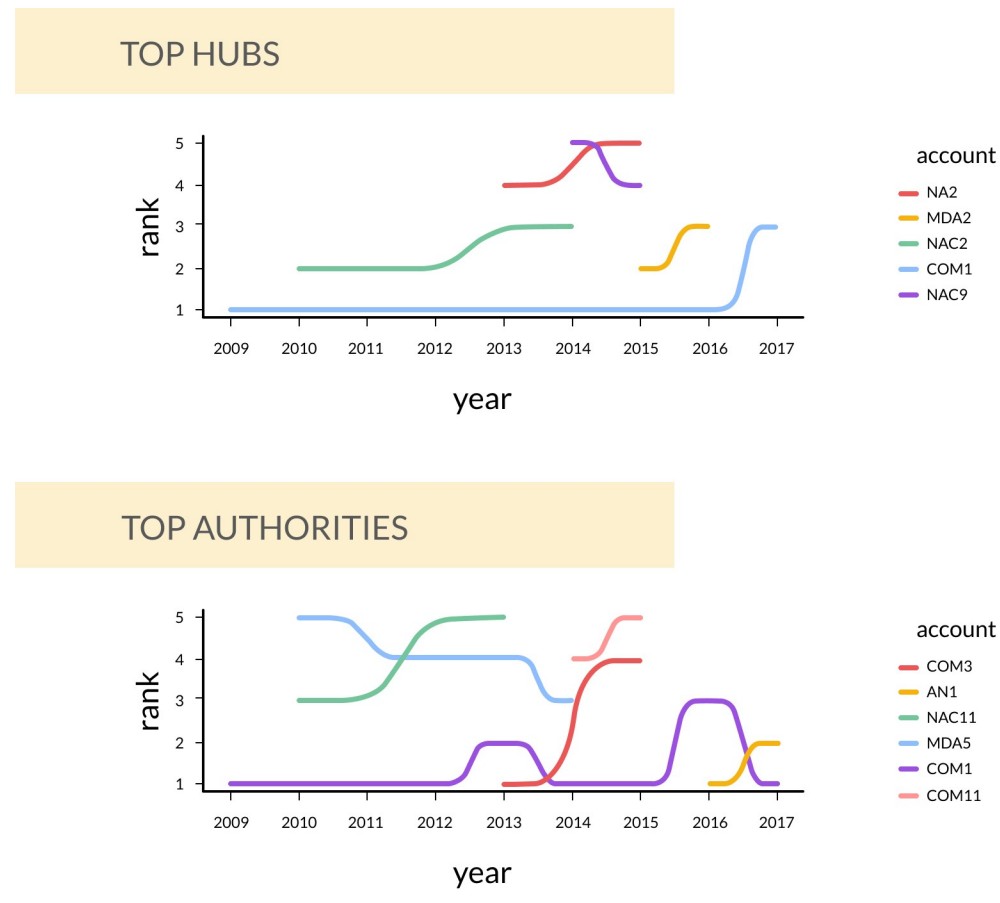

**Fig 4. Bump charts demonstrating longitudinal patterns in top hubs and authorities of Twitter probiotics chatter from 2009–2017.** Lower numbers on the y-axis indicate a higher rank. COM1 is the only account that is consistently present in both the top hubs amd authorities charts. COM (Commerical Organizations) are more prevalent as top authorities while having a minimal presence as top hubs.

### Posting and tagging correlation of probiotic Twitter accounts (RQ4)

In Fig 5, the in-degree (x-axis) and out-degree (y-axis) values of probiotic Twitter accounts are plotted in scatter plots for the years 2009–2017. The top two accounts with highest degree values are labelled in these plots. In addition, the percentage of accounts which have out-degree values more than in-degree values are displayed alongside the year in the plots. For the first four years (2009–2012), there are more accounts with higher in-degree values than out-degree values since the accounts with higher out-degree percentage is below 50%. However, the next five years (2013–2017) indicate an opposite trend with majority of the accounts having higher out-degree than in-degree. This indicates tweet posting propensity is more than being tagged in tweets since 2013. The scatterplots also show that COM1 has been a consistent influencer in the probiotic Twitter networks by maintaining a balance between posting and tagging.

## Discussion

Consumer interest in probiotics products as measured through online searchers has grown from 2004–19 [43]. Recognizing this trend, e-commerce is now a priority for the probiotics

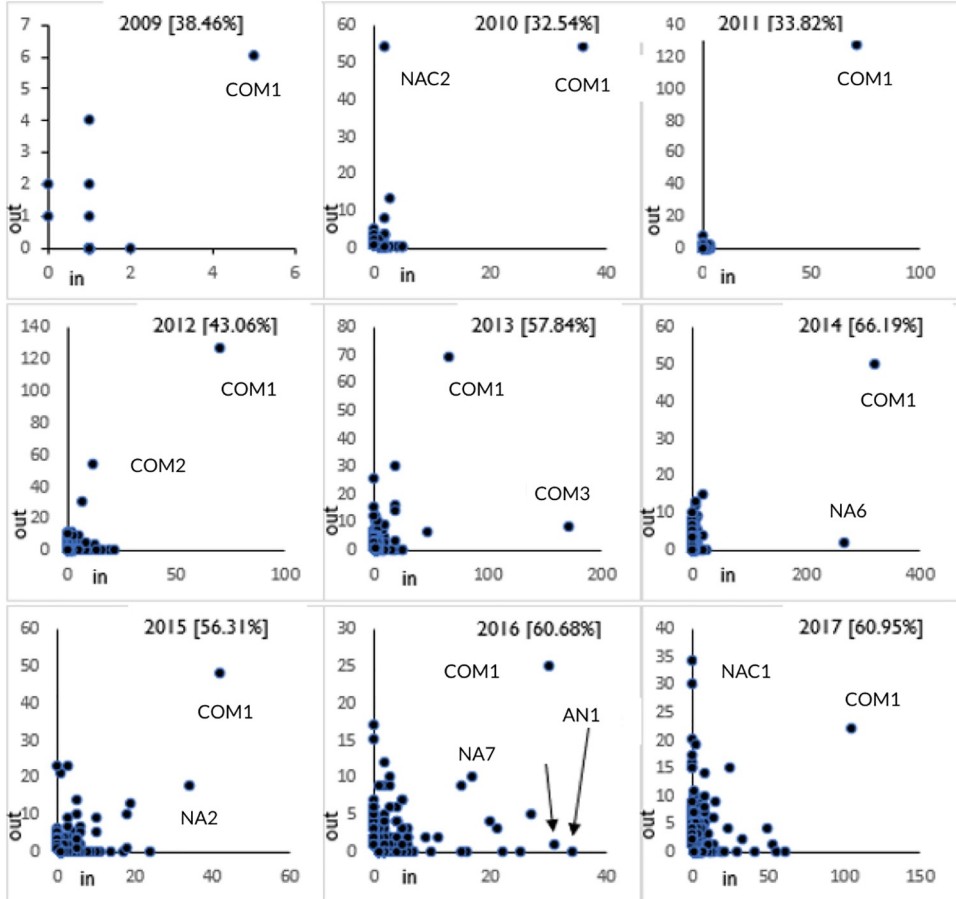

**Fig 5. Plots of in-degree values (posting) against out-degree values (tagging) demonstrating online behaviour for key influencers are plotted in scatter plots for the years 2009–2017.** The top two accounts with highest degree values are labelled in these plots. In addition, the percentage of accounts which have out-degree values more than in-degree values are displayed alongside the year in the plots. COM (Commerical Organizations) consistently appear as top accounts across the years. The tagging behavior is dominant until 2012 while posting behavior takes precedence in the last five years of the analysis period.

industry which leverages market intelligence tools and resources to (a) better understand and monitor online engagement trends more efficiently, and (b) reach consumers for better yields around specific product categories or formats. These trends are worthy of attention from the perspectives of public health nutrition and dietetics professionals should they choose to intervene in the informational environments of popular nutritional products such as probiotics whose health efficacy continues to be debated. It is in this context that our study identifies key actors in the online probiotics network in Twitter over time, quantifies their level of influence, and documents shifts in probiotics communities over a nine-year period from 2009–2017. Our longitudinal social network analyses offers several novel findings that merit discussion.

Discussions on all ten topics by consumers indicate positive connotations of words to health, ranging from health promotion (e.g. "good", "health", "healthy") to treatment of disease or health conditions (e.g. "help", "treatment", "patient"). Topic 0 reflects consumer discussions that indicate their association between probiotics and health, consistent with the established awareness and consumer understanding of the benefits of probiotics [44].

Topic 1 shows consumer confidence in probiotics as being new and of the belief that it can be used for treatment. Topic 2 relates to consumers' discussions on connotations to health

claims, with words such as "may", "could", "study" and "reduce" having higher weight relative to their frequency. This confirms that consumers view food beyond providing taste, aroma and basic nutritional needs to seeing probiotics as a form of functional food that provides additional physiological benefit targeting at improving consumers' health and wellness [45].

Topics 3 and 9 indicate consumers' association of probiotics with food. In Topic 3, the term "kefir" has significant higher weight relative to the frequency. This is in line with the increased interest in health benefits and microbial composition of on kefir as a potential product containing probiotics warranting further research [46]. Similarly, this trend is seen in Topic 9 with the highest weight relative to word frequency discussed attributed to "super," followed by "cheese".

Topic 7 points towards consumers' discussions on probiotics and market growth. This trend is also consistent with data that shows that the global probiotics market has experienced tremendous growth at more than USD44.2 billion in 2019 and is projected to rise at a compounded annual growth rate of 7.7% by the end of 2025 with consumers consuming more probiotics with awareness for a healthy diet and its nutritious content [47].

In terms of the social network analysis findings, the steady growth in the number of probiotics communities from 2009 reveals rising consumer and advertiser interest in them. From 2010, we notice a larger diversity in social network activity spurred by the emergence of new players and a variety of emerging communities. This trend reaches its peak until 2014 after which period the network tends to saturate towards an equilibrium but at a heightened level of communication activity as compared to its genesis in 2009. While the number of communities might have stabilised, the denser network graph in 2017 indicates heightened tweeting activity and a larger number of accounts who used the term 'probiotics'.

However, it is evident that not all communities have experienced an equal level of sustenance or success. Specifically, we find that COM1, a probiotics company in the UK, is the only actor that has maintained a consistent position as the leading hub and authority in the UK probiotics Twitter network across the nine year period. A closer look at the statistics suggests that their investment into outreach in the initial years, 2009–2012, might have reaped returns from 2013–17 positioning them also as the main authority from 2013–2017. We observe that a majority of the other actors who have been assigned one of the top five ranks are commercial entities as opposed to individuals, suggesting that individual influencer effects or involvement in the UK Twitter probiotics network might be minimal.

The interlinkages between the dynamics of content and communities can be best understood by analysing Fig 1 (that identifies prevalent themes) in the context of Fig 3, which visualises the movement in communities across the nine years. While the top three models (Models 0, 1 and 2) suggest that Twitter chatter around probiotic products has been dominated by their characterisation as functional foods and the health benefits they offer, Fig 3 demonstrates how conversations around these health benefits have preoccupied online conversations across the study period and culminated with a surge in 2017. These findings are resonant with Burges-Watson, Moreira and Murtagh's [48] qualitative observations about the "ambigious promise" of probiotic products where the benefits portrayed in popular representation such as advertising are "incommensurate" with scientific evidence. The main inference we draw from this finding, in concert with the predominance of online advertising in our dataset, is that the online information ecosystem of probiotic products might have experienced shifts in volume of chatter, but have remained largely consistent in terms of content.

From a nutrition education perspective, these findings suggest that scientists studying the health effects of probiotics supplements, governmental agencies or regulators that oversee controversial labelling issues around probiotic products, or science journalists who play a critical role in disseminating scientific news around probiotics to the public exerted minimal level of influence in these networks during this period. This trend was finally bucked in 2016 when the

AN1, a professional association of dieticians appeared in the network to swiftly emerge as a top-ranked authority and maintained its position in the following year despite minimal out-reach (they do not appear ranked for out-degree scores).

Finally, our mapping of communities in the social network suggests that there has been consistent rise in multi-brand advertising and the promotion of health benefits of probiotic products. These findings find resonance in the work of Brinich and colleagues (2013) who suggested that patients might harbour unrealistic expectations of probiotic products should they read content on probiotic websites that singularly highlight its therapeutic benefits [10]. Also relevant to this discussion is the discursive analysis of probiotics websites which promoted these products as being essential to one's vitality strategically situated within the larger issue of the individual being responsible for their own health [49]. We observe that communities discussing the health effects of probiotics from a critical standpoint appear sporadically for a relatively brief shelf-life as compared to other communities that are geared towards advertising and different kinds of promotional strategies.

Our findings bear implications for communication strategies aimed at creating a more balanced information ecosystem about probiotic products. Specifically, apart from a few exceptions (AC 1–4) the community of probiotic scientists is clearly underrepresented on Twitter and weild minimal influence on the probiotics chatter. Twitter can be valuable to scientists in terms of disseminating their science to non-scientific audiences and engage with policymakers as well; both affordances which are of high relevance to the probiotics context [50] given the prevailing power of advertising and labelling controversies surrounding probiotic products. Scientists can forge new networks of communication [51] with non-academic users who have been shown in our study to weild influence in probiotics communities on Twitter. Lastly, our study demonstrates that scientists and nutrition policymakers may tag professional organisations like AN1 who, despite their seemingly limited following, may be developing growing influence in probiotics-related Twitter communities. Essentially, scientists and policymakers may imbibe the approach of commercial organisations whose efforts to grow as a Twitter authority seems to have been built on the efforts of being a hub of probiotics-related communication.

The generalizability of our findings is constrained by four main methodological limitations. First, by considering only tweets that contain the terms 'probiotic' or 'probiotics', our analysis could be missing other relevant tweets which do not contain these terms but might still be related to issues surrounding probiotics or probiotic supplement. Our rationale for adopting the approach we did was to use terms that would offer us both, the specificity and breadth to be able to capture the dataset of most relevance to our research questions. Second, the twitter graphs built for this study are not a representation of the standard graph-theoretic model. It is to be highlighted that we are interpreting the in-degree and out-degree values as proxy measures for tagging and posting behavior of user accounts (nodes) in the graph. Third, the analysis of tweets for identifying community theme names, could be more robust if independent coding of the tweets was conducted. However, the large number of tweets rendered this process time-consuming. Accordingly, the review and confirmation of the themes from a second coder was sought as an acceptable compromise. Finally, we analyzed data from Twitter for this study. However, users may have used other social media platforms such as Facebook and Reddit to discuss about probiotics. Thus, this study's findings may not fully represent the overall social media discussion on probiotics.

## Conclusion

Using probiotics as an exemplar of a nutritional issue characterized by conflicting information, our study longitudinally chronicles the evolution, growth, and decline of virtual communities

related to this functional food product in the context of Twitter. We discovered a predominance of commercial entities over time and the relatively limited influence of non-commercial, academic, regulator or media-related actors in these networks. These findings suggest that should these trends remain consistent we may expect to see an asymmetrical online informational environment around probiotics products focused on promoting its benefits and attracting consumers using a range of promotional strategies. In the context of conflicting, equivocal evidence around probiotics, it is incumbent upon allied stakeholders such as scientists, media, and policymakers to engage with these communities with an aim to minimize consumer confusion. Given the expanding remit of probiotics-related e-commerce, future research may expand the scope of this study by focusing on other social media and online platforms where consumers engage in conversations around food, diet and nutrition.

## Supporting information

**S1 Fig. Plots of the top ten key words in each topic superimposed onto the weight the model places on such words in each topic.** In a similar vein to PCA, higher weights equal more importance in the model. As a general rule, the frequency of the word should not significantly exceed the weight. Words that have a higher frequency relative to the weight are often less important.
(TIF)

## Author Contributions

**Conceptualization:** Santosh Vijaykumar, Aravind Sesagiri Raamkumar, Kristofor McCarty.

**Data curation:** Cuthbert Mutumbwa.

**Formal analysis:** Aravind Sesagiri Raamkumar, Kristofor McCarty, Cyndy Au.

**Funding acquisition:** Santosh Vijaykumar.

**Methodology:** Aravind Sesagiri Raamkumar, Kristofor McCarty.

**Project administration:** Santosh Vijaykumar.

**Resources:** Santosh Vijaykumar.

**Software:** Santosh Vijaykumar, Cuthbert Mutumbwa.

**Supervision:** Santosh Vijaykumar.

**Writing – original draft:** Santosh Vijaykumar, Aravind Sesagiri Raamkumar, Kristofor McCarty, Jawwad Mustafa.

**Writing – review & editing:** Santosh Vijaykumar.

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
