## [Decision Letter · Decision Letter 0]

27 Apr 2021

PONE-D-21-05002

Themes, Communities and Influencers of Online Probiotics Chatter: A Retrospective Analysis from 2009-2017

PLOS ONE

Dear Dr. Vijaykumar,

Thank you for submitting your manuscript to PLOS ONE. After careful consideration, we feel that it has merit but does not fully meet PLOS ONE’s publication criteria as it currently stands. Therefore, we invite you to submit a revised version of the manuscript that addresses the points raised during the review process.

Please carefully consider the reviewers' comments. The statistical rigor of the paper should be improved in the revised version. Please justify the need for the word clouds as they do not typically convey information in a rigorous way. Please provide adequate descriptions for the rest of the figures or consider changing them in order to make them easier to understand.

We look forward to receiving your revised manuscript.

Kind regards,

Liviu-Adrian Cotfas

Academic Editor

PLOS ONE

Journal Requirements:

2. Please confirm that the study conformed to all terms and conditions of the websites used.

4. Thank you for stating the following in the Financial Disclosure section:

'This project was funded by the Consumer Data Research Centre, an ESRC Data Investment, under project ID CDRC 085, ES/L011840/1; ES/L011891/1. The grant was awarded to SV (PI) and KM (Co-I). The funders had no role in study design, data collection and analysis, decision to publish, or preparation of the manuscript.'

We note that one or more of the authors are employed by a commercial company: Martel Instruments

Reviewers' comments:

Reviewer's Responses to Questions

**Comments to the Author**

1. Is the manuscript technically sound, and do the data support the conclusions?

Reviewer #1: Partly

Reviewer #2: Yes

Reviewer #3: Yes

2. Has the statistical analysis been performed appropriately and rigorously? 

Reviewer #1: No

Reviewer #2: Yes

Reviewer #3: Yes

3. Have the authors made all data underlying the findings in their manuscript fully available?

Reviewer #1: No

Reviewer #2: Yes

Reviewer #3: Yes

4. Is the manuscript presented in an intelligible fashion and written in standard English?

Reviewer #1: Yes

Reviewer #2: Yes

Reviewer #3: Yes

5. Review Comments to the Author

Reviewer #1: The authors have attempted to understand the interplay between content and community dynamics via online probiotics chatter, by applying topic modeling and social network analysis methods to their empirical analysis. It has been common interest to uncover evolving processes of topic-sensitive online communities across diverse research fields. The manuscript handles a significant topic, but its approaches are not sportive enough to address the topic. There are several concerns that the author needs to consider and revise accordingly.

1. Methods

Using Gephi, the authors conducted community detection in a giant component each year and show the basic statistics such as #nodes, #edges, and #communities in Table 1. The authors should consider diverse aspects of network properties to discuss time-evolving online communities. Only sizes of giant components and the numbers of communities are not enough to show your community dynamics. Also, ForceAtlas2 (line#215-216) is not a community detection algorithm but a graph layout algorithm.

Additionally, out- and in-degrees of a node only exhibit the shallow depth of a network (i.e. immediate neighbors), which is not enough to show influencers of a whole network. The authors are recommended to apply more comprehensive ways to figure out influencers, referring to the state-of-the-art approaches.

2. Unclear findings

This study is to analyze retrospective online probiotics chatter, but the contributions and outcomes are not quite clear. The authors kindly introduce background of probiotics, but the interlinked aspects of probiotics communities, in terms of content dynamics and community dynamics, are not clear.

3. Figures

Overall, figures do not help readers understand main messages. The authors need to consider different ways of showing their results with detailed captions.

Reviewer #2: The topic and approach is very interesting. It would be better if authors clearly described the method. It is not clear if this is a mixed method approach. Additionally, they used thematic analysis on collected tweets and it would be helpful if they described the code generation steps (how much level of agreement between coders) and showed the final codebook in a table. I am also wondering if there are sub-categories / sub-communities within each community / category. This might provide higher level of granularity.

Reviewer #3: This is an interesting and sophisticated analysis of Twitter discussion relating to probiotics from 2009 to 2017. Extant Python tools are used to identify prominent accounts, subnetworks and topics within the wider network. Of particular interest is whether Twitter users have access to a balanced perspective (certainly an important class of question in this time of polarising echo chambers).

I hope that this can be published; beyond the interest of the findings themselves, the tools and methods will certainly be of interest to readers. In fact, with that in view, I think it would be valuable to see a bit more detail about the implementation, or perhaps to make the code used available in supplementary materials. I think it could benefit from some other revisions too. My main concern is that it's not really clearly shown how the more sophisticated analysis has given us better understanding. In the Conclusion, the actionable finding seems to be just that commercial interests are dominating online discussion of probiotics. Couldn't a much simpler analysis have told us that? Could this deeper analysis lead to specific strategic information about HOW to provide balance, e.g., by identifying the communities most in need of a scientific perspective, or identifying who public health communicators should tag to most effectively disseminate info?

I have answered "Yes" when asked "Have the authors made all data underlying the findings in their manuscript fully available?" although actually all I can see is that they have stated an intention to do so, via the Open Science Foundation repository. This would need to be actioned prior to publication.

Also I have answered "yes" regarding the clarity of the English, although there is room for improvement. I believe I've mentioned most of the worst offences below, but there were a few other minor typos; please do another proofread.

Smaller comments follow.

58: You refer to the favourable information as "these claims" here, but the first two sentences seem to assert them as fact. It might be more consistent if you changed "that confer" in line 54 to "that reputedly confer" or something like that.

67: I'm not sure what's meant by "expectedly." Did someone predict that this would happen?

92 (and similarly later): The words "let," "allow" and "enable" are synonymous, but "let" takes the bare infinitive while "allow" and "enable" take the full infinitive. Thus you can say "allow us to understand" or "let us understand," but not "let us to understand" or "allow us understand," and "enable" works the same way as "allow."

105: "comprising of," while apparently fine in Singapore, is incorrect in international English; see, e.g., https://en.wiktionary.org/wiki/comprise#cite_note-1.

111: misplaced hyphen

111-115: These definitions should be more specific; since the in-degree and out-degree are important in this study, the reader should know exactly what they are. Does "actively communicating to other members of the network" require tagging? Do you literally mean "whether," or is this a count? It shouldn't necessarily require any more text than this to specify exactly what these quantities are.

169: Twitter handles are anonymised here, but later, in the Results section, we see a number of the most prominent accounts by name. I'm sure this is all legitimate (and the tweets are public, after all) but adding a few more words might clarify the matter.

211-217: This should be a little clearer about whether there is one giant component or a number of them (e.g., is there some threshold size, or is the giant component simply the largest connected component?), and also about what ForceAtlas2 is doing to detect communities (conceptually, if a precise description is prohibitive).

243: Table 2

245: "sence" should be "sense." More importantly, it would be good to see an example of this topic overlap.

246: "ultimaley"

248: This section is unclear. Its heading refers to RQ2, but the final sentence says that it "pertains to RQ1." Is there a missing "also," or is this an error? (Similar statements appear at the end of the next two sections too.) Also, what are "coders" in this context? Are they people? If so, did they view a random sample of tweets from each community, or what exactly? Did the second coder independently assign themes before comparing with the first coder's choices?

253: This is a conditional clause, not a whole sentence.

268: This sentence needs rewriting for grammar and clarity.

Figure 1: Now that I reach the figures, I note that (at least as they have reached me) the resolution is much too low. I'm sure the journal has its own process to ensure that hi-res figures are used in the final version, but as it is, I can't even read the smaller words in Fig 1. I know these word-clouds are popular, but printing them at sufficient size to make the smallest words legible will require quite a lot of space; I would consider a table.

284: "the word clouds in Figure 4"? Figure 1, right?

283-285: So you're saying that the words used to define each topic tend to appear in the tweets attributed to that topic, right? That sounds inevitable, prima facie. Does this serve as a check against something that could have gone wrong? I guess what I'm saying is that it would be good to clarify this significance, if any, of the agreement between weights and frequencies within each topic.

291: "neighbouring" should be "neighbour"

297: delete a 'most'

298: transposed L and E in 'conversely'

290-299: The discussion of Figure 2 is reasonable as far as it goes, but the Figure has some striking features that aren't mentioned and may disquiet the reader. In particular, the tweets seem mostly gathered into filiform structures. It's conceivably an illusion due to the dimension reduction required for the plot, but the default assumption would be that these apparent structures are meaningful. It's therefore surprising to see individual structures split up, partly allocated to one topic & partly to another. For instance, right near the top of the plot, we see five or six contiguous threads coloured partly red and partly yellow. Could a different choice of algorithm parameters in the topic allocation phase provide better agreement between the imposed partition and the finer partition that appears to be visible in this plot? Or perhaps you can offer an interpretation.

322-330: One thing the alluvial plot shows that's not commented on here is the movement, from year to year, of Twitter accounts between different communities. It would be good to see some comment about that. While some amount of movement is probably natural (as individual account holders' interests develop), there does seem to be a surprising amount here. Does that raise red flags for the robustness of the model? Was year-to-year consistency of community membership a criterion in the community allocations, or if not, could it have been?

343: since all the other Twitter accounts are in lowercase and preceded by an "@", "Optibac" here should be too, to avoid confusion.

383-387: If I understand the degrees correctly, then every time one account tags another, one out-degree increases by 1 and one in-degree increases by 1, i.e., the sum of the out-degrees is the sum of the in-degrees. Therefore, interpreting the proportion of accounts in which in-degree exceeds out-degree is a delicate matter, reflecting I suppose a comparison of the skews of the in-degree and out-degree distributions. I don't really understand the sentence on line 387, but I don't think it's a fair interpretation of this rather indirect observation.

401: dangling modifier

6. PLOS authors have the option to publish the peer review history of their article (what does this mean?). If published, this will include your full peer review and any attached files.

Reviewer #1: No

Reviewer #2: **Yes: **Sherif Abdelhamid

Reviewer #3: **Yes: **Dr David W Bulger

---

## [Author Response · Author response to Decision Letter 0]

14 Jul 2021

The responses to all reviewers are included in the file titled Response to Reviewers that has been uploaded with the rest of the documents.

---

## [Decision Letter · Decision Letter 1]

3 Aug 2021

PONE-D-21-05002R1

Themes, Communities and Influencers of Online Probiotics Chatter: A Retrospective Analysis from 2009-2017

PLOS ONE

Dear Dr. Vijaykumar,

Thank you for submitting your manuscript to PLOS ONE. After careful consideration, we feel that it has merit but does not fully meet PLOS ONE’s publication criteria as it currently stands. Therefore, we invite you to submit a revised version of the manuscript that addresses the points raised during the review process.

In the revised version of the manuscript you are invited to consider the reviewers' comments included below, such as providing more meaningful captions for the figures and adding additional details.

We look forward to receiving your revised manuscript.

Kind regards,

Liviu-Adrian Cotfas

Academic Editor

PLOS ONE

Journal Requirements:

Reviewers' comments:

Reviewer's Responses to Questions

**Comments to the Author**

1. If the authors have adequately addressed your comments raised in a previous round of review and you feel that this manuscript is now acceptable for publication, you may indicate that here to bypass the “Comments to the Author” section, enter your conflict of interest statement in the “Confidential to Editor” section, and submit your "Accept" recommendation.

Reviewer #1: (No Response)

Reviewer #2: All comments have been addressed

Reviewer #3: (No Response)

2. Is the manuscript technically sound, and do the data support the conclusions?

Reviewer #1: Yes

Reviewer #2: Yes

Reviewer #3: Partly

3. Has the statistical analysis been performed appropriately and rigorously? 

Reviewer #1: Yes

Reviewer #2: Yes

Reviewer #3: Yes

4. Have the authors made all data underlying the findings in their manuscript fully available?

Reviewer #1: Yes

Reviewer #2: Yes

Reviewer #3: Yes

5. Is the manuscript presented in an intelligible fashion and written in standard English?

Reviewer #1: Yes

Reviewer #2: Yes

Reviewer #3: Yes

6. Review Comments to the Author

Reviewer #1: The authors have improved the manuscript by incorporating the reviewers’ suggestions. There are some more recommendations that the authors need to consider as follows.

1) Table 1 needs to show standard deviations as well as the averages of network properties for more accurate interpretations of evolving community structures, since measured properties of all nodes unlikely follow normal distributions in the real world.

2) The authors are recommended to interpret the interlinked aspects of probiotic communities between dynamics of content and communities using their results from analysis. The paragraphs four to eight in the intro are too general to understand the interlinked dynamics of this study.

3) I could not find any changes in figure captions. Figure captions need to be self-explanatory without having to read detailed content in the manuscript. Also, figure legends and resolutions are too small and low to recognize.

Reviewer #2: The authors clearly responded to most of my reviews and comments and I think this manuscript is now acceptable for publication.

Reviewer #3: I think the article is substantially improved, and most of my concerns have been addressed. I am particularly persuaded by the new discussion on Figure 2.

There is one place in particular where the authors have chosen to respond to my and the other reviewer's concerns by providing explanations in the author response, without altering the manuscript. That seems odd. If the reviewers require that explanation, then almost certainly some section of the readership would as well. I am thinking especially of the coding method. I think it's worth clarifying this more in the manuscript.

The addition of "closeness centrality" may be a step in the right direction, but is currently unclear. Neither a definition nor a citation for this statistic is provided, and the interpretation is confusing. It is introduced as "a way of detecting nodes that are able to spread information very efficiently," and yet it seems to be computed at either the cluster or population level rather than the node level. Thus it's mysterious how such a measure could identify which nodes spread info efficiently, and generally unclear what it's measuring.

Lastly, the in-degree vs out-degree count is not resolved to my satisfaction. In-degrees and out-degrees are indeed familiar from graph theory (though I think the revised explanation could be useful for part of the readership). However, in a directed graph, the total in-degree and total out-degree ARE equal. The authors seem to be comparing the number of tweets tagging an account with the number of tweets posted by an account, but in order to see those as in-degree and out-degree, we'd need to have ONE edge emerging from a tweet, and then splitting (away from any node) into MULTIPLE edges to terminate at each of the tagged accounts. This is not a standard graph-theoretic model; it's dubious that it's the best way to handle it, but certainly it's unreasonable to use this model without explaining it.

7. PLOS authors have the option to publish the peer review history of their article (what does this mean?). If published, this will include your full peer review and any attached files.

Reviewer #1: No

Reviewer #2: No

Reviewer #3: **Yes: **David Bulger

---

## [Author Response · Author response to Decision Letter 1]

14 Sep 2021

Dear Editor,

Warm greetings. At the outset, I would like to thank you for facilitating the second round of reviews. We are gratified with the overall positive responses to the major and minor changes we have made to the manuscript and are thankful to the reviewers for helping to further refine the manuscript.

In concert with the reviewers’ comments, we have now made the following changes: 1) included standard deviations for the average degree and average closeness centrality metrics, 2) discussed interlinkages between content and communities, 3) revised captions to make them more self-explanatory, 4) included descriptions of the coding process in the manuscript with additional detail, 5) provided fuller explanations of the closeness centrality measure, and 6) acknowledged the issue of in-degree vs. out-degree count in the limitations section.

We hope that these changes are satisfactory to you and the reviewers and look forward to your feedback to this revised version of our manuscript.

Best wishes,

Santosh Vijaykumar (on behalf of all authors)

Response to Reviewers

Reviewer # 1: The authors have improved the manuscript by incorporating the reviewers’ suggestions. There are some more recommendations that the authors need to consider as follows.

Table 1 needs to show standard deviations as well as the averages of network properties for more accurate interpretations of evolving community structures, since measured properties of all nodes unlikely follow normal distributions in the real world.

Authors Response: Thanks for the suggestion. As requested, we have added standard deviations for the average degree and average closeness centrality metrics in Table 1. 

2) The authors are recommended to interpret the interlinked aspects of probiotic communities between dynamics of content and communities using their results from analysis. 

Authors Response: We have now added an interpretation of the interlinked dynamics of content and communities through the following paragraph:

The interlinkages between the dynamics of content and communities can be best understood by analysing Figure 1 (that identifies prevalent themes) in the context of Figure 3, which visualises the movement in communities across the nine years. While the top three models (Models 0, 1 and 2) suggest that Twitter chatter around probiotic products has been dominated by their characterisation as functional foods and the health benefits they offer, Figure 3 demonstrates how conversations around these health benefits have preoccupied online conversations across the study period and culminated with a surge in 2017. These findings are resonant with Burges-Watson, Moreira and Murtagh’s 40 qualitative observations about the “ambigious promise” of probiotic products where the benefits portrayed in popular representation such as advertising are “incommensurate” with scientific evidence. The main inference we draw from this finding, in concert with the predominance of online advertising in our dataset, is that the online information ecosystem of probiotic products might have experienced shifts in volume of chatter but have remained largely consistent in terms of content. 

Reviewer: The paragraphs four to eight in the intro are too general to understand the interlinked dynamics of this study.

Authors Response: We thank the reviewer for their feedback but have reviewed this portion and believe that the content is specific to the probiotics context. Having said so, we would be happy to consider suggestions for the inclusion of any specific literature or perspectives in this section provided by the reviewer.

3) I could not find any changes in figure captions. Figure captions need to be self-explanatory without having to read detailed content in the manuscript. Also, figure legends and resolutions are too small and low to recognize.

Authors Response: We have now amended the captions for all figures and hope the revised versions are satisfactory. We have formatted figure lends and resolutions in concordance with the journal’s requirements but are happy to make any specific changes we might be requested.

Reviewer #2: The authors clearly responded to most of my reviews and comments and I think this manuscript is now acceptable for publication.

Authors Response: Thank you very much for your positive response and the constructive reviews.

Reviewer #3: I think the article is substantially improved, and most of my concerns have been addressed. I am particularly persuaded by the new discussion on Figure 2.There is one place in particular where the authors have chosen to respond to my and the other reviewer's concerns by providing explanations in the author response, without altering the manuscript. That seems odd. If the reviewers require that explanation, then almost certainly some section of the readership would as well. I am thinking especially of the coding method. I think it's worth clarifying this more in the manuscript.

Authors Response: Thanks for highlighting this issue. We went over the comments that we had submitted for the previous revision, pertaining to the coding exercise. We noticed some missing points, which have now been added to the section titled “Identification of Communities (RQ2)” in Page 12. Here is the full excerpt – 

“As mentioned earlier, 53 communities were identified in the Twitter graphs built for the nine-year time period. The prevalent theme of community was identified using a process involving one principal coder and one reviewer (two of the authors). First, the principal coder reviewed the contents of randomly selected tweets from each community and assigned the relevant theme names.38 Next, another author reviewed and confirmed the themes assigned by the principal coder. It is to noted that this author did not independently assign the themes to the tweets, rather reviewed the assignments of the first coder. If a community had more than one major theme, the community’s label was set to theme1 and theme2. For each of these themes, the number of constituent communities are reported. We did not use any codebook for this exercise.”

The addition of "closeness centrality" may be a step in the right direction, but is currently unclear. Neither a definition nor a citation for this statistic is provided, and the interpretation is confusing. It is introduced as "a way of detecting nodes that are able to spread information very efficiently," and yet it seems to be computed at either the cluster or population level rather than the node level. Thus it's mysterious how such a measure could identify which nodes spread info efficiently, and generally unclear what it's measuring.

Authors Response: The closeness centrality measure in Gephi is an implementation of a corresponding algorithm which was first proposed in a research paper [1]. As per Gephi’s wiki page, the development team define the metric as “the average distance from a given node to all other nodes in the network”. As per our understanding, the metric is calculated for each node in the graph. We acknowledge that the sentences in the previous manuscript pertaining to this metric, do appear inadequate. Accordingly, we have made some suitable amendments. Please find the excerpt below.

“The average closeness centrality [1] per node, has been included in Table 1. This metric defines the importance of a node in the graph, by measuring how close the node is to other nodes in the graph (sum of geodesic distance between the particular node and all other nodes in the graph). In a graph of multiple nodes, the nodes with relatively lower closeness centrality values, are considered to be closer to the other nodes in the graph. With 2009 as the exception, it is observed that this metric has consistently increased every year at an average level. Although probiotic Twitter users seem to be posting more tweets through the years, the proximity to each other has been steadily decreasing as indicated by the rise in average closeness centrality values.”

[1] Brandes, Ulrik. "A faster algorithm for betweenness centrality." Journal of mathematical sociology 25.2 (2001): 163-177.

Lastly, the in-degree vs out-degree count is not resolved to my satisfaction. In-degrees and out-degrees are indeed familiar from graph theory (though I think the revised explanation could be useful for part of the readership). However, in a directed graph, the total in-degree and total out-degree ARE equal. The authors seem to be comparing the number of tweets tagging an account with the number of tweets posted by an account, but in order to see those as in-degree and out-degree, we'd need to have ONE edge emerging from a tweet, and then splitting (away from any node) into MULTIPLE edges to terminate at each of the tagged accounts. This is not a standard graph-theoretic model; it's dubious that it's the best way to handle it, but certainly it's unreasonable to use this model without explaining it.

Authors Response: Thank you for the comment. Yes, you are right that the total in-degree of all nodes in a graph should match with the total out-degree. We would like to clarify that our comment in the earlier response letter, pertained only to singular nodes for which the in-degree and out-degree vary but at a total graph level, these metrics add up to the same value. We also acknowledge that the twitter graph representation in this paper, is not the standard graph-theoretic model. This is a limitation in this study. We have included this point a limitation in the paper. The limitations paragraph has been inserted below.

“The generalizability of our findings is constrained by four main methodological limitations. First, by considering only tweets that contain the terms ‘probiotic’ or ‘probiotics’, our analysis could be missing other relevant tweets which do not contain these terms but might still be related to issues surrounding probiotics or probiotic supplement. Our rationale for adopting the approach we did was to use terms that would offer us both, the specificity and breadth to be able to capture the dataset of most relevance to our research questions. Second, the twitter graphs built for this study are not a representation of the standard graph-theoretic model. It is to be highlighted that we are interpreting the in-degree and out-degree values as proxy measures for tagging and posting behavior of user accounts (nodes) in the graph. Third, the analysis of tweets for identifying community theme names, could be more robust if independent coding of the tweets was conducted. However, the large number of tweets rendered this process time-consuming. Accordingly, the review and confirmation of the themes from a second coder was sought as an acceptable compromise. Finally, we analyzed data from Twitter for this study. However, users may have used other social media platforms such as Facebook and Reddit to discuss about probiotics. Thus, this study’s findings may not fully represent the overall social media discussion on probiotics.”

---

## [Editor Report · Decision Letter 2]

20 Sep 2021

Themes, Communities and Influencers of Online Probiotics Chatter: A Retrospective Analysis from 2009-2017

PONE-D-21-05002R2

Dear Dr. Vijaykumar,

We’re pleased to inform you that your manuscript has been judged scientifically suitable for publication and will be formally accepted for publication once it meets all outstanding technical requirements.

Kind regards,

Liviu-Adrian Cotfas

Academic Editor

PLOS ONE
---

## [Editor Report · Acceptance letter]

8 Oct 2021

PONE-D-21-05002R2 

Themes, Communities and Influencers of Online Probiotics Chatter: A Retrospective Analysis from 2009-2017 

Dear Dr. Vijaykumar:

I'm pleased to inform you that your manuscript has been deemed suitable for publication in PLOS ONE. Congratulations! Your manuscript is now with our production department. 

Kind regards, 

on behalf of

Dr. Liviu-Adrian Cotfas 

Academic Editor

PLOS ONE